# Association of SFG *Rickettsia massiliae* and *Candidatus* Rickettsia shennongii with Different Hard Ticks Infesting Livestock Hosts

**DOI:** 10.3390/pathogens12091080

**Published:** 2023-08-24

**Authors:** Shehla Shehla, Farman Ullah, Abdulaziz Alouffi, Mashal M. Almutairi, Zaibullah Khan, Tetsuya Tanaka, Marcelo B. Labruna, Kun-Hsien Tsai, Abid Ali

**Affiliations:** 1Department of Zoology, Abdul Wali Khan University Mardan, Mardan 23200, Pakistan; shehla@awkum.edu.pk (S.S.); zaib.zoologist@gmail.com (Z.K.); 2King Abdulaziz City for Science and Technology, Riyadh 12354, Saudi Arabia; 3Department of Pharmacology and Toxicology, College of Pharmacy, King Saud University, Riyadh 11451, Saudi Arabia; 4Laboratory of Infectious Diseases, Joint Faculty of Veterinary Medicine, Kagoshima University, Kagoshima 890-0065, Japan; k6199431@kadai.jp; 5Department of Preventive Veterinary Medicine and Animal Health, Faculty of Veterinary Medicine, University of São Paulo, São Paulo 05508-060, Brazil; labruna@usp.br; 6Institute of Environmental and Occupational Health Sciences, Department of Public Health, College of Public Health, National Taiwan University, Taipei 100025, Taiwan

**Keywords:** ticks, *Rickettsia massiliae*, *Candidatus* Rickettsia shennongii, livestock, Pakistan

## Abstract

Ixodid ticks are responsible for the transmission of various intracellular bacteria, such as the *Rickettsia* species. Little Information is available about the genetic characterization and epidemiology of *Rickettsia* spp. The current study was designed to assess the tick species infesting various livestock hosts and the associated *Rickettsia* spp. in Pakistan. Ticks were collected from different livestock hosts (equids, cattle, buffaloes, sheep, goats, and camels); morphologically identified; and screened for the genetic characterization of *Rickettsia* spp. by the amplification of partial fragments of the *gltA*, *ompA* and *ompB* genes. Altogether, 707 ticks were collected from 373 infested hosts out of 575 observed hosts. The infested hosts comprised 105 cattle, 71 buffaloes, 70 sheep, 60 goats, 34 camels, and 33 equids. The overall occurrence of *Rickettsia* spp. was 7.6% (25/330) in the tested ticks. *Rickettsia* DNA was detected in *Rhipicephalus haemaphysaloides* (9/50, 18.0%), followed by *Rhipicephalus turanicus* (13/99, 13.1%), *Haemaphysalis cornupunctata* (1/18, 5.5%), and *Rhipicephalus microplus* (2/49, 4.1%); however, no rickettsial DNA was detected in *Hyalomma anatolicum* (71), *Hyalomma dromedarii* (35), and *Haemaphysalis sulcata* (8). Two *Rickettsia* agents were identified based on partial *gltA*, *ompA*, and *ompB* DNA sequences. The *Rickettsia* species detected in *Rh. haemaphysaloides*, *Rh. turanicus*, and *Rh. microplus* showed 99–100% identity with *Rickettsia* sp. and *Candidatus* Rickettsia shennongii, and in the phylogenetic trees clustered with the corresponding *Rickettsia* spp. The *Rickettsia* species detected in *Rh. haemaphysaloides*, *Rh. turanicus*, *Rh. microplus*, and *Ha. cornupunctata* showed 100% identity with *R. massiliae*, and in the phylogenetic trees it was clustered with the same species. *Candidatus* R. shennongii was characterized for the first time in *Rh. haemaphysaloides*, *Rh. turanicus*, and *Rh. microplus*. The presence of SFG *Rickettsia* spp., including the human pathogen *R. massiliae*, indicates a zoonotic risk in the study region, thus stressing the need for regular surveillance.

## 1. Introduction

Ticks (Acari, Parasitiformes, Ixodida) are blood-feeding ectoparasites found in all ecoregions of the world [1]. They are known for transmitting different pathogens, including viruses, protozoans, and bacteria, that pose significant threats to human and animal health [2,3]. Among tick-borne pathogens, species of the bacterial genera *Anaplasma*, *Ehrlichia*, and *Rickettsia*, as well as protozoans of the genera *Babesia* and *Theileria*, cause infection in small ruminants and bovines [4,5], which may lead to significant economic losses. These pathogens also cause infections in humans in many regions [6,7,8]. In addition, many tick-borne *Rickettsia* spp. play a relevant public role by causing spotted fever illness in different parts of the world [4].

There have been significant advances in our knowledge of the diversity and distribution of *Rickettsiae* in many parts of world [4,9]. Recently, the characterization of novel *Rickettsia* spp. based on reliable genetic markers revealed broad diversity at the species level [9,10]. Currently, the genus *Rickettsia* encompasses 34 recognized species plus several unidentified strains [11]. These species are divided into five main monophyletic groups: the spotted fever group (SFG), the transitional group, the typhus group, the canadensis group, and the bellii group [11]. Species belonging to these groups differ in terms of several traits related to their ecology, distribution, hosts, and pathogenicity [4]. The SFG is highly diverse and a subject of intensive study [9], encompassing the largest number of tick-borne pathogens, including *Rickettsia africae*, *Rickettsia conorii*, *Rickettsia honei*, *Rickettsia japonica*, *Rickettsia massiliae*, *Rickettsia monacensis*, *Rickettsia parkeri*, *Rickettsia rickettsii*, *Rickettsia sibirica*, and *Rickettsia slovaca* [12].

*Rickettsia massiliae* was first reported in *Rhipicephalus sanguineus* ticks infesting dogs from France [13]. Since then, the agent has been detected in different tick species, including *Rhipicephalus microplus*, *Rhipicephalus turanicus*, *Rhipicephalus haemaphysaloides*, *Rh. sanguineus*, *Rhipicephalus lunulatus*, *Rhipicephalus pusillus*, *Rhipicephalus bursa*, *Rhipicephalus sulcatus*, *Rhipicephalus muhsamae*, *Haemaphysalis punctata*, *Haemaphysalis erinacei*, *Haemaphysalis parva*, *Haemaphysalis adleri*, *Haemaphysalis sulcata*, *Dermacentor marginatus*, *Ixodes ricinus*, and *Hyalomma anatolicum* in several countries of Africa, Europe, and the Middle East [2,3,4,14,15]. There have also been some reports of *R. massiliae* in *Rh. sanguineus* sensu lato (s.l.) in the New World [4,16,17,18]. Furthermore, *R. massiliae* has been molecularly characterized in *Ha. bispinosa*, *Rh. sanguineus*, and *Rh. turanicus* ticks in India, Iran, and China, respectively [19,20,21].

Pakistan has a great variety of landscapes and habitats, maintaining a broad range of vertebrate species which serve as hosts for diverse tick species and reservoirs for different types of pathogens, particularly *Rickettsia* spp. [3,22,23,24,25,26,27,28]. Earlier, serological methods were used to detect the exposure of humans to rickettsial infection in Pakistan [29], but antigen conservation among different species makes it difficult to accurately identify and explore the existing and novel diversity of *Rickettsia* spp. [5]. In Pakistan, few molecular studies have been conducted for the detection and characterization of *Rickettsia* spp. [5,30], although there have been recent records of *R. massiliae* in *Hy. anatolicum*, *Hyalomma hussaini*, *Rh. haemaphysaloides*, *Rh. turanicus*, and *Rh. microplus* ticks [2,3,8]. Indeed, currently available data regarding the existing of several unidentified *Rickettsia* spp. in Pakistan are scarce, as the available records have been confined to small sampling areas and limited host ranges, and based on only one or two genetic markers. Therefore, the present study aimed to use three suitable genetic markers (*gltA*, *ompA*, and *ompB*) for the identification and genetic characterization of *Rickettsia* spp. in ticks infesting diverse hosts in different agro-ecological zones of Pakistan.

## 2. Materials and Methods

### 2.1. Study Area

Tick specimens were collected in the following 13 districts of Khyber Pakhtunkhwa (KP) province, Pakistan: Charsadda (34.161297° N, 71.753660° E), Bajaur (34.7865° N, 71.5249° E), Swabi (34.1241° N, 72.4613° E), Mardan (34.194697° N, 72.050557° E), Peshawar (34.039825° N, 71.566832° E), Bannu (32.9298° N, 70.6693° E), Lower Dir (34.9161° N, 71.8097° E), Malakand (34.5030° N, 71.9046° E), Lakki Marwat (32.6135° N, 70.9012° E), Nowshera (34.0105° N, 71.9876° E), Mohmand (34.5356° N, 71.2874° E), Swat (34.8065° N, 72.3548° E), and Buner (34.3943° N, 72.6151° E). The KP province has suitable environmental conditions for different ticks and tick-borne pathogens because of its desertic, humid, and arid plains and the arid and humid hilly areas that vary in climate, altitude, and seasons (winter, spring, summer, and fall). The average temperatures of the selected districts of KP province range from 33.4 °C in the summer to 10.4 °C in the winter (climate-data.org, accessed on 7 April 2023). These areas are characterized by an abundance of free-roaming hosts infested with ticks and close coexistence of humans and animals. To design a map, the geographic coordinates of all collection sites were obtained using a Global Positioning System, imported into a Microsoft Excel sheet, and processed using ArcGIS V. 10.3.1 (ESRI, Redlands, CA, USA) (Figure 1).

### 2.2. Tick Sampling and Identification

All of the ticks (males, females, and nymphs) were conveniently collected during July 2019 to October 2020 from different livestock hosts (equids, cattle, buffaloes, sheep, goats, and camels) in the 13 districts. Ticks were collected from the aforementioned hosts whenever they were found, irrespective of specific location within the targeted survey regions and the time, in various farms, open fields, and freely moving animals in pastures. Collection was carried out once from each host when found to be infested with ticks. By examining the entire body of each host, 1–8 attached ticks per animal were collected using tweezers. Immediately after collection, the ticks were washed with distilled water followed by 70% ethanol and stored in properly labeled tubes containing 100% ethanol. Collected ticks were morphologically identified under a stereo zoom microscope (SZ61, Olympus, Tokyo, Japan) using standard taxonomic keys [31,32,33,34].

### 2.3. DNA Extraction and PCR

The collected ticks were individually subjected to DNA extraction using the standard phenol chloroform method [35]. The genomic DNA was extracted from 330 selected ticks (118 nymphs, 95 males, and 117 females). The extracted DNA was quantified via NanoDrop (NanoQ, Optizen, Daejeon, Republic of Korea). The pellet was hydrated with nuclease-free water. A conventional PCR (GE-96G, BIOER, Hangzhou, China) was performed to amplify partial fragments of the rickettsial citrate synthase (*gltA*), 190-kDa outer membrane protein (*ompA*), and 120-kDa outer membrane protein (*ompB*) genes (Table 1). PCR assays were performed in a 20 µL reaction volume containing 12 µL PCR master mix (Thermo fisher scientific, Inc.; Waltham, MA, USA), 1 µL of each forward and reverse primer (10 µM), 2 µL of genomic DNA (50–100 ng), and 4 µL of PCR water. The thermo-cycling conditions for the amplification of the *gltA*, *ompA*, and *ompB* genes were used as described previously [36,37,38]. *Rickettsia aeschlimannii* DNA and nuclease-free water were used as positive and negative controls, respectively. The PCR products were run on 1.5% agarose gel prepared in tris borate EDTA (TBE) containing 2 µL ethidium bromide at a concentration of 0.2–0.5 μg/mL for staining purposes. The amplified DNA fragments were observed by means of gel documentation (BioDoc-It™ Imaging Systems UVP, LLC, Upland, CA, USA).

### 2.4. DNA Sequencing and Phylogenetic Analysis

All of the PCR-amplified products for the *gltA*, *ompA*, and *ompB* genes were submitted for bidirectional sequencing (Macrogen, Inc., Seoul, Republic of Korea). The obtained sequences were trimmed to remove the poor reading sequences through SeqMan v. 5 (DNASTAR, Inc.; Madison, WI, USA), and additionally to generate partial sequences for *gltA*, *ompA*, and *ompB* genes. The obtained sequences were submitted to BLAST (Basic Local Alignment Search Tool) [39] at NCBI (National Center for Biotechnology Information). For the construction of phylogenetic trees, *Rickettsia* spp. sequences were retrieved from GenBank and aligned with the obtained sequences using the BioEdit Sequence Alignment Editor v. 7.0.5 [40]. Individual phylogenetic trees based on the *gltA*, *ompA*, and *ompB* fragments were constructed in MEGA-X [41] using the maximum likelihood method and the Tamura–Nei model [42]. All the available methods were tested, being found similar results. However, the maximum likelihood is a recommended and accurate method for the best evolutionary analysis, due to its ability to evaluate different phylogenetic trees and models under a statistical framework [43]. Moreover, the topology of *Rickettsia* spp. in this MS was in accordance to Karkouri et al. [11]. Bootstrap resampling analysis (1000 replicates) was used to assess the statistical significance of the nodes. The final positions in the dataset comprised the obtained *gltA*, *ompA*, and *ompB* fragments.

## 3. Results

### 3.1. Occurrence of Morphologically Identified Ticks

Overall, 707 ticks were collected from 373 (64.9%) out of 575 examined livestock hosts. The tick-infested animals consisted of 105 cattle, 71 buffaloes, 70 sheep, 60 goats, 34 camels, and 33 equids. The highest number of infestation rates of the different hosts was recorded in the of district Nowshera (29/35, 82.9%), followed by Buner (33/40, 82.5%), Swabi (32/39, 82.1%), Swat (29/39, 74.3%), Bajaur (20/27, 74.1%), Lakki Marwat (28/39, 71.8%), Lower Dir (26/39, 66.7%), Peshawar (35/58, 60.3%), Bannu (21/35, 60.0%), Mohmand (23/39, 58.9%), Mardan (29/52, 55.7%), Charsadda (49/89, 55.0%), and Malakand (19/44, 43.1%). The highest occurrence of ticks was recorded in the district of Charsadda (91/707, 12.9%) followed by Peshawar (73/707, 10.3%), Mardan (70/707, 9.9%), Lakki Marwat (62/707, 8.8%), Swabi (57/707, 8.06%), Nowshera (57/707, 8.1%), Bajaur (53/707, 7.5%), Buner (53/707, 7.5%), Swat (49/707, 6.9%), Bannu (42/707, 5.9%), Mohmand (37/707, 5.2%), Lower Dir (34/707, 4.8%), and Malakand (29/707, 4%). Based on morphological analyses, seven tick species belonging to three genera (*Rhipicephalus*, *Haemaphysalis*, and *Hyalomma*) were identified. Overall, the highest occurrence was observed for *Rh. microplus* (179/707, 25.3%), followed by *Rh. turanicus* (163/707, 23.1%), *Hy. anatolicum* (135/707, 19.09%), *Rh. haemaphysaloides* (119/707, 16.8%), *Hyalomma dromedarii* (74/707, 10.5%), *Haemaphysalis cornupunctata* (28/707, 3.9%), and *Ha. sulcata* (9/707, 1.3%). Among the 707 collected ticks, 320 (45.3%) were females, 219 (30.9%) were nymphs, and 168 (23.8%) were males.

### 3.2. Detection of Rickettsia spp. in Ticks

Table 2 describes the results of molecular analyses of 330 ticks (118 nymphs, 95 males, and 117 females) collected from different hosts in different locations, which were individually subjected to amplification of fragments of each of the rickettsial genes *gltA*, *ompA*, and *ompB*. The overall occurrence of *Rickettsia* spp. based on *gltA*, *ompA*, and *ompB* genes was 7.6% (25/330) of the tested ticks. *Rickettsia* spp. were detected in four tick species: *Rh. haemaphysaloides*, *Rh. turanicus*, *Rh. microplus*, and *Ha. cornupunctata*, while no *Rickettsia* sp. was detected in *Hy. anatolicum*, *Hy. dromedarii*, and *Ha. sulcata*. The occurrence of *Rickettsia* spp. was highest in *Rhipicephalus haemaphysaloides* (9/50, 18.0%), followed by *Rh. turanicus* (13/99, 13.1%), *Ha. cornupunctata* (1/18, 5.5%), and *Rh. microplus* (2/49, 4.1%). The occurrence of *Rickettsia* spp. was noted to be highest in the district of Mardan (4/26, 15.4%), followed by Mohmand (2/20, 10.0%), Bajaur (3/33, 9.1%), Charsadda (3/34, 8.8%), Lower Dir (2/23, 8.7%), Nowshera (2/26, 7.7%), Swabi (2/28, 7.1%), Peshawar (2/30, 6.7%), Malakand (1/18, 5.5%), Swat (1/19, 5.3%), Bannu (1/20, 5.0%), Lakki Marwat (1/26, 3.8%), and Buner (1/27, 3.7%) (Table 2).

### 3.3. Sequencing and Phylogenetic Analysis

DNA sequences were generated from all 25 tick specimens that yielded amplicons by means of PCR assays targeting fragments of the rickettsial genes *gltA*, *ompA*, and *ompB*. Two *Rickettsia* spp. agents were identified based on partial fragments of *gltA*, *ompA*, and *ompB*; i.e., *Candidatus* Rickettsia shennongii and *R. massiliae* (Table 2). For each of these two *Rickettsia* spp., a single haplotype was generated for each of the three rickettsial genes, regardless of the number of PCR-positive ticks. The obtained *gltA*, *ompA*, and *ompB* haplotypes from *Rh. haemaphysaloides*, *Rh. turanicus*, and *Rh. microplus* showed maximum identities of 100%, 99.29–99.82%, and 99.87–100% and queries of 71–100%, 99%, and 94%, respectively, with *Rickettsia* sp. and *Ca*. R. shennongii, which have been reported in Taiwan and China. On the other hand, a rickettsial *gltA* haplotype obtained from *Rh. haemaphysaloides*, *Rh. turanicus*, *Rh. microplus*, and *Ha. cornupunctata* showed 100% identity and 100% query with *R. massiliae* reported in the USA, whereas an *ompA* haplotype from these same ticks showed 99.3–100% maximum identity and 100% query with *R. massiliae* reported in USA, France, and Lebanon, and an *ompB* haplotype showed 99.24–100% maximum identity and 100% query with *R. massiliae* reported in Argentina.

Since only one haplotype was generated for each gene of *Ca*. R. shennongii and *R. massiliae*, this single haplotype was used in each of the phylogenetic analyses. In all three phylogenetic trees, the obtained *Ca*. R. shennongii haplotypes clustered with the *Rickettsia* sp. and *Ca*. R. shennongii on the basis of *gltA*, *ompA*, and *ompB* reported from Taiwan and China, respectively (Figure 2, Figure 3 and Figure 4). The *R. massiliae* sequences clustered with the same species based on *gltA* reported from the USA; *ompA* reported from the USA, France, and Lebanon; and *ompB* reported from the USA, France, and Argentina (Figure 2, Figure 3 and Figure 4). The obtained haplotypes of *Ca*. R. shennongii were deposited to GenBank under the following accession numbers: *gltA* (OP820487, OR428237–OR428243, OR437436–OR437448), *ompA* (OP820485, OR437460–OR437479), and *ompB* (OP820483, OR437449–OR437459). The *R. massiliae* haplotypes were deposited under the following accession numbers: *gltA* (OP820488, OR428235, OR428236), *ompA* (OP820486, OR428231), and *ompB* (OP820484, OR428232–OR428234).

## 4. Discussion

Prior to this study, the SFG novel agent *Ca*. R. shennongii was detected in *Rh. haemaphysaloides* ticks in China [12], while *R. massiliae* was detected in different tick species in different parts of the world. The pathogenic role of *R. massiliae* has been reported in humans [6,44,45], while the pathogenicity of *Ca*. R. shennongii to humans is unknown. The ecological conditions in Pakistan are suitable for the propagations of ticks and tick-borne pathogens [2,5]. Previous studies reporting *R. massiliae* in ticks from Pakistan relied on, at most, two genetic markers [2,3,5]. However, due to the unavailability of sufficient knowledge regarding the unidentified *Rickettsia* spp. in Pakistan, there is a need to conduct comprehensive studies in which *Rickettsia* spp. could be genetically characterized via suitable genetic markers. Hence, the present study reported two SFG *Rickettsia* spp. in four tick species via *gltA*, *ompA*, and *ompB* genetic markers. In phylogenetic trees inferred from *gltA*, *ompA*, and *ompB* partial sequences, the present sequences of *Ca*. R. shennongii and *R. massiliae* grouped separately with their corresponding species from different regions. The *R. massiliae* sequences were grouped into two branches, suggesting evolutionary differences.

The highest tick occurrence was noted for *Rh. microplus*, which is a dominant tick in the region [2,3,46]. Equids were found to be infested by *Rh. turanicus* and *Rh. haemaphysaloides*. The ticks *Rh. microplus*, *Haemaphysalis bispinosa*, *Hy. anatolicum*, and *Hy. dromedarii*, previously reported when found on Pakistan’s equids [2,46], were not found on the host species of the present study. This may be due to the availability of other suitable hosts. A wide host range was observed for *Rh. turanicus*, *Rh. haemaphysaloides*, *Hy. anatolicum*, *Hy. dromedarii*, *Ha. cornupunctata*, and *Ha. sulcata*, infesting cattle, buffaloes, sheep and goats; this might be due to their three-host life cycle [47]. The one-host tick *Rh. microplus*, infesting various hosts such as cattle, buffaloes, and goats, may be linked to the sharing of habitats by different hosts. The camels were found to be infested by *Hy. dromedarii*, which is considered the main tick species parasitizing camels [48].

*Rhipicephalus turanicus*, *Rh. microplus*, *Rh. haemaphysaloides*, and *Ha. cornupunctata* ticks infesting cattle, buffaloes, sheep, and goats were found to be positive for *R. massiliae*. Previously, *R. massiliae* has been detected in tick species of the genera *Rhipicephalus*, *Haemaphysalis*, *Hyalomma*, and *Dermacentor*, collected from dogs, small ruminants, and cattle in China, Iran, Nigeria, Tunisia, Portugal, Argentina, and the USA [17,49,50,51,52,53,54,55]. *Rickettsia massiliae* in *Rh. microplus*, *Rh. turanicus*, and *Rh. haemaphysaloides* ticks infesting cattle, buffaloes, goats, sheep, and equids has been reported in Pakistan [2,5,8], although this is the first report of *R. massiliae* in *Ha. cornupunctata* infesting sheep and goats. In previous studies, the highest occurrence of *R. massiliae* was observed in *Rh. microplus* and *Rh. haemaphysaloides* [2,3], while in the present study, the highest occurrence was observed in *Rh. haemaphysaloides*. This may be due the collection process of tick samples from the hosts (equids and wild animals), which were different than the hosts examined in the current study (livestock).

The agent *Ca*. R. shennongii was detected in three tick species—*Rh. microplus*, *Rh. haemaphysaloides*, and *Rh. turanicus*—infesting cattle, buffaloes, sheep, and goats. Previously, this *Rickettsia* species was not fully characterized and mostly called *Rickettsia* sp., and has been detected in *Haemaphysalis spinigera*, *Haemaphysalis turturis*, *Haemaphysalis bandicota*, and *Rh. haemaphysaloides* reported from India (NCBI https://www.ncbi.nlm.nih.gov/, accessed on 7 April 2023) and Taiwan [56]. In some cases, it was called *R. massiliae* when detected in ectoparasites of pets reported from India (NCBI https://www.ncbi.nlm.nih.gov/, accessed on 7 April 2023). Recently, it was genetically characterized and called *Ca*. R. shennongii when detected in *Rh. haemaphysaloides* ticks in China [12], confirming its broad host and geographic range. This also provides evidence for the possible role of these ticks in the spreading of *Ca*. R. shennongii, as the adult female and nymph ticks were found to be positive for rickettsial DNA. Hence, there is a possibility that the detected rickettsial DNA was ingested in blood from the infected host. *Rickettsia* spp.-infected ticks constitute a possible health risk to livestock-holders [4], and we stress the need for further research to understand its pathogenic potential and to avoid any zoonotic consequences.

The *gltA*, *ompA*, and *ompB* genes have been shown to have a high degree of intraspecific variation, and are extensively used for reliable phylogenetic analyses within the genus *Rickettsia* [57]. Taking these into account, the characterization of *Ca*. R. shennongii and *R. massiliae* was confirmed by these three reliable genetic markers [2,3,8]. Based on these genetic markers, the sequences of *Ca*. R. shennongii showed maximum identities with sequences of *Rickettsia* sp. and “*Ca*. R. shennongii” reported in Taiwan and China [12], while the sequences of *R. massiliae* showed maximum identities with sequences of this species reported in the USA and Argentina. The pathogenic potential of *R. massiliae* in humans has been described in Europe and South America [6,46]; however, there is no information about the pathogenicity of *Ca*. R. shennongii. Although the zoonotic transmission of *R. massiliae* in Pakistan is currently unknown, it emphasizes the need to conduct further epidemiological studies in order to explore its pathogenic role. The systematic investigation of SFG *Rickettsia* spp. with high zoonotic potential [58] may allow us to explore the emerging novel species in the region. The presence of *Ca*. R. shennongii in Pakistan indicates the presence of diverse unidentified SFG *Rickettsia* spp. Extensive “One-Health” studies and various surveillance programs are essential in order to elucidate the epidemiology, transmission, and pathogenicity of *Rickettsia* spp. in the country. The One-Health approach is particularly relevant for the development of strategies to control tick infestations and associated TBDs. The integration of the One-Health approach in surveillance programs will improve our understanding regarding the circulation of zoonotic TBPs in different regions of the country.

## 5. Conclusions

The present study reports the presence of two *Rickettsia* agents in *Rh. haemaphysaloides*, *Rh. turanicus*, *Rh. microplus*, and *Ha. cornupunctata*, collected from cattle, buffaloes, sheep, and goats, in 13 districts of KP, Pakistan: *Ca*. R. shennongii associated with *Rh. haemaphysaloides*, *Rh. microplus*, and *Rh. turanicus*; and *R. massiliae* associated with *Rh. haemaphysaloides*, *Rh. turanicus*, *Rh. microplus*, and *Ha. cornupunctata*. The distribution of *Rickettsia* in the study area and the observed detection rate in domestic animals point to the diversity of SFG *Rickettsiae*. Epidemiological and surveillance studies are required in order to explore the pathogenic potential of *Ca*. R. shennongii.

## Figures and Tables

**Figure 1 pathogens-12-01080-f001:**
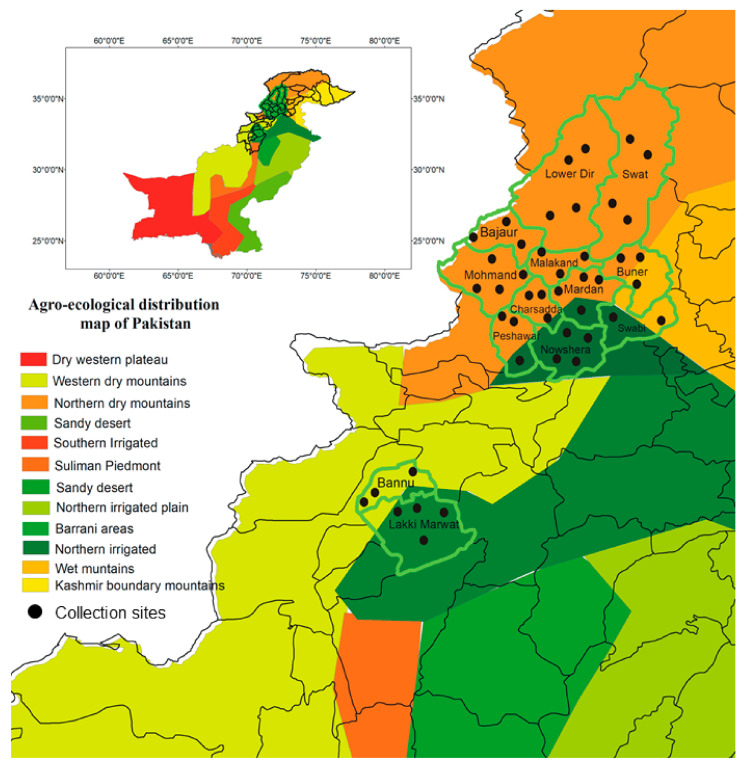
Map showing collection sites for ticks in different districts of Khyber Pakhtunkhwa, Pakistan.

**Figure 2 pathogens-12-01080-f002:**
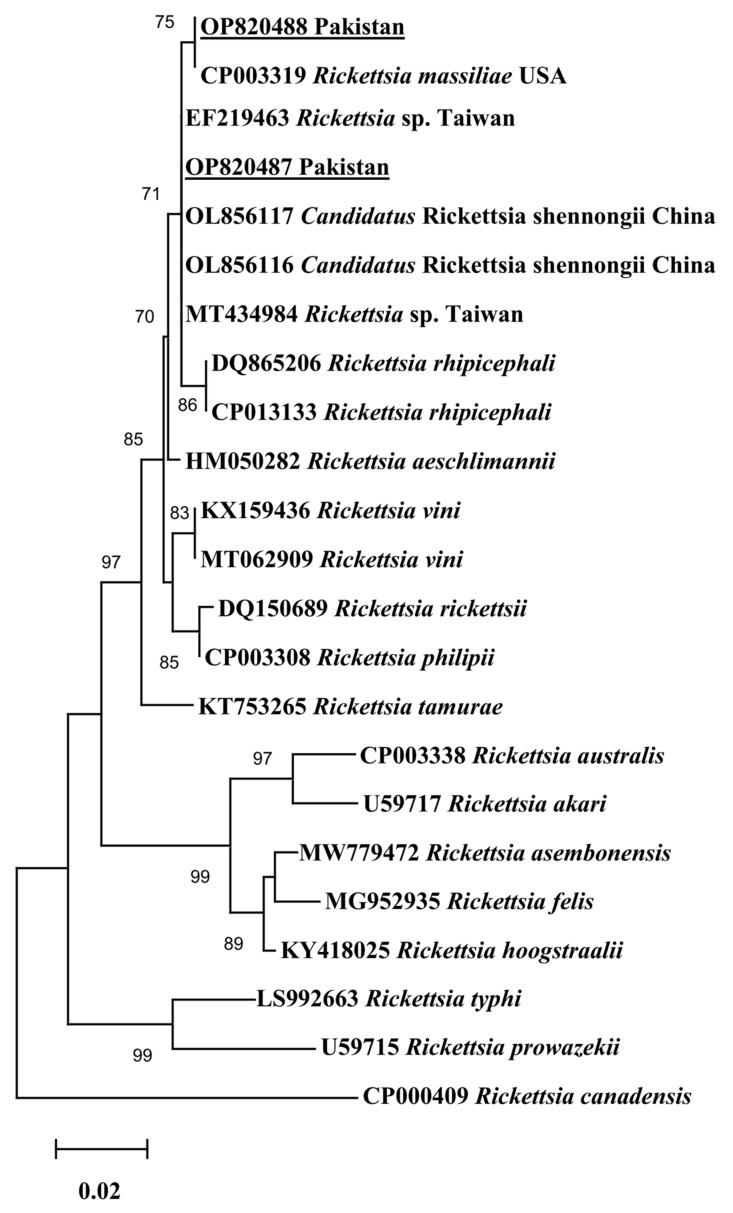
Phylogenetic tree based on *gltA* sequences for *Rickettsia* spp. using the Maximum likelihood method (Tamura–Nei model). The *Rickettsia canadensis* was taken as an outgroup. The 1000 bootstrapping values were followed, and the levels of bootstrap support (≥70%) for the phylogenetic groupings are given at each node. The sequences (OP820487; 363 bp, OP820488; 362 bp) obtained in the present study are shown in an underlined font.

**Figure 3 pathogens-12-01080-f003:**
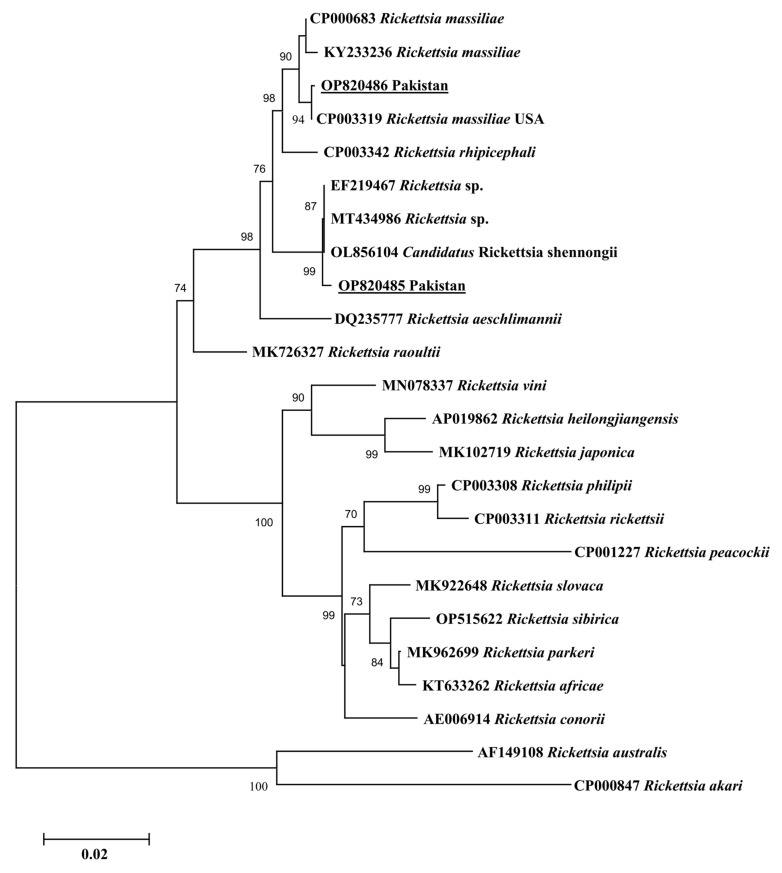
Phylogenetic tree based on *ompA* sequences for *Rickettsia* spp. using the Maximum likelihood method (Tamura–Nei model). The group comprising sequences of *Rickettsia akari*, and *Rickettsia australis* was taken as the outgroup. The 1000 bootstrapping values were followed, and the levels of bootstrap support (≥70%) for the phylogenetic groupings are given at each node. The sequences (OP820485; 564 bp, OP820486; 530 bp) obtained in the present study are shown in an underlined font.

**Figure 4 pathogens-12-01080-f004:**
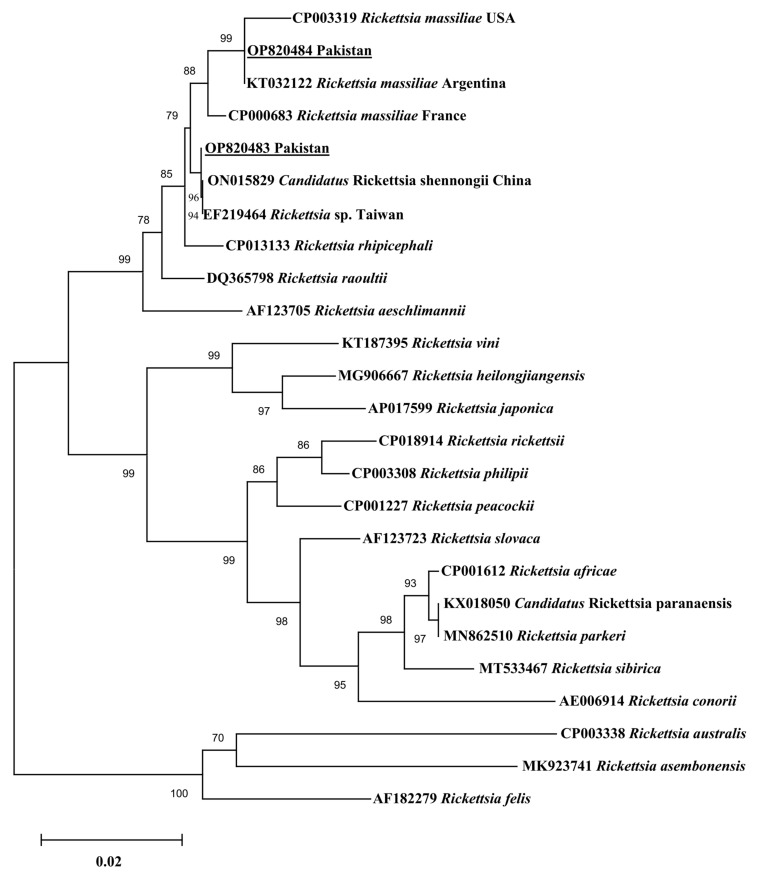
Phylogenetic tree based on *ompB* sequences for *Rickettsia* spp. using the Maximum likelihood method (Tamura–Nei model). The group comprising sequences of *Rickettsia australis*, *Rickettsia asembonensis*, and *Rickettsia felis* was taken as the outgroup. The 1000 bootstrapping values were followed, and the levels of bootstrap support (≥70%) for the phylogenetic groupings are given at each node. The sequences (OP820483; 796 bp, OP820484; 794 bp) obtained in the present study are shown in an underlined font.

**Table 1 pathogens-12-01080-t001:** List of primers used in the present study.

Gene	Primer	Sequence (5′-3′)	Amplicon Size	Reference
*gltA*	CS-78	GCAAGTATCGGTGAGGATGTAAT	401 bp	[36]
	CS-323	GCTTCCTTAAAATTCAATAAATCAGGAT		
*ompA*	Rr190.70p	ATGGCGAATATTTCTCCAAAA	631 bp	[37]
	Rr190.701n	GTTCCGTTAATGGCAGCATCT		
*ompB*	120-M59	CCGCAGGGTTGGTAACTGC	862 bp	[38]
	120-807	CCTTTTAGATTACCGCCTAA		

**Table 2 pathogens-12-01080-t002:** Occurrence of ticks and the detection rate of *Rickettsia* spp. According to geographical districts and hosts in Pakistan.

District	Hosts	No. of Infested/No. of Examined Livestock Hosts	Identified Ticks	No. of Ticks According to Life Stage	No. of Tested Ticks	Molecular Detection of *Rickettsia* spp. In Morphologically Identified Ticks (N, F, M)					
						No. of Ticks with *R. massiliae*			No. of Ticks with *Ca.* R. shennongii		
						*gltA*	*ompA*	*ompB*	*gltA*	*ompA*	*ompB*
Charsadda	Equids	4/9	*Rh. turanicus*	1N, 2F	1N, 1F				1N	1N	1N
			*Rh. haemaphysaloides*	3N, 1M, 1F	1N, 1M, 1F						
	Cattle	14/20	*Rh. turanicus*	1N, 1M, 2F	1M, 1F						
			*Hy. anatolicum*	2N, 2F	1N, 1F						
			*Rh. haemaphysaloides*	4N, 1M, 3F	1N, 1M, 1F			1F			
			*Rh. microplus*	7N, 2M, 11F	1N, 1M, 1F						
	Buffaloes	8/15	*Hy. anatolicum*	1N, 1M, 3F	1N, 1M, 1F						
			*Rh. turanicus*	1N, 1M, 1F	1M, 1F				1F	1F	
	Sheep	8/20	*Hy. anatolicum*	1N, 1M, 2F	1N, 1F						
			*Rh. turanicus*	2N, 1M, 2F	1M						
	Goats	9/15	*Hy. anatolicum*	1N, 1M, 1F	1N, 1F						
			*Rh. turanicus*	2N, 1M, 3F	1N, 1M, 1F						
			*Rh. haemaphysaloides*	3N, 2M, 6F	1N, 1M, 1F						
	Camels	6/10	*Hy. dromedarii*	3N, 2M, 5F	1N, 1M, 1F						
	Total	49/89		91	34	3					
Peshawar	Equids	4/11	*Rh. turanicus*	2N, 1M, 2F	1N, 1M						
	Cattle	13/19	*Hy. anatolicum*	4N, 2M, 5F	1N, 1M, 1F						
			*Rh. haemaphysaloides*	2N, 2M, 4F	1N, 1M, 1F						
	Buffaloes	5/9	*Hy. anatolicum*	2N, 1M, 3F	1N, 1M, 1F						
			*Rh. turanicus*	3N, 1M, 3F	1N, 1M, 1F						
	Sheep	6/8	*Rh. turanicus*	3N, 1M, 3F	1N, 1M, 1F						
	Goats	5/7	*Rh. haemaphysaloides*	4N, 1M, 4F	1N, 1M, 1F				1N	1N	1N
			*Rh. microplus*	3N, 2M, 6F	1N, 1M, 1F				1N	1N	1N
	Camels	2/4	*Hy. dromedarii*	3N, 1M, 5F	1N, 1M, 5F						
	Total	35/58		73	30	2					
Mardan	Equids	3/5	*Rh. turanicus*	1N, 2M, 5F	1N, 1M, 1F				1M	1M	
	Cattle	11/15	*Hy. anatolicum*	1N, 2M, 2F	1N, 1M, 1F						
			*Rh. turanicus*	1N, 2M, 3F	1N, 1M, 1F	1N	1N	1N			
			*Rh. microplus*	7N, 4M, 13F	1N, 1M, 1F						
	Buffaloes	5/10	*Rh. haemaphysaloides*	2N, 2M, 3F	1N, 1M, 1F						
	Sheep	4/10	*Rh. haemaphysaloides*	2N, 2M, 2F	1N, 1M, 1F				1N	1N	
	Goats	4/8	*Rh. haemaphysaloides*	1N, 2M	1N, 1M						
			*Rh. turanicus*	3N, 2M, 1F	1N, 1M, 1F				1N	1N	1N
	Camels	2/4	*Hy. dromedarii*	1N, 2M, 2F	1N, 1M, 1F						
	Total	29/52		70	26	4					
Swabi	Equids	3/5	*Rh. turanicus*	1N, 2M, 1F	1N, 1M, 1F						
	Cattle	9/9	*Rh. microplus*	5N, 3M, 7F	3N, 1M, 1F						
	Buffaloes	8/8	*Hy. anatolicum*	1N, 2M, 3F	1N, 1M, 1F						
			*Rh. turanicus*	1N, 1M, 1F	1N, 1M, 1F				1N	1N	1N
	Sheep	5/7	*Rh. haemaphysaloides*	1N, 2M, 3F	1N, 1M, 1F						
			*Rh. turanicus*	1N, 1M, 2F	1N, 1M, 1F						
	Goats	5/6	*Rh. microplus*	2N, 3M, 5F	1N, 2M, 2F	1M		1M			
	Camels	2/4	*Hy. dromedarii*	2N, 3M, 4F	1N, 1M, 1F						
	Total	32/39		57	28	2					
Lakki Marwat	Equids	3/4	*Rh. turanicus*	1N, 1M, 2F	1N, 1M, 1F						
	Cattle	7/8	*Hy. tnatolicum*	1N, 1M, 1F	1N, 1M, 1F						
			*Rh. turanicus*	3N, 1M, 1F	1N, 1F						
			*Rh. microplus*	3N, 2M, 6F	1N, 2M						
	Buffaloes	5/7	*Hy. anatolicum*	1N, 2M, 3F	1N, 1F						
	Sheep	6/10	*Hy. anatolicum*	3N, 1M, 4F	1N, 1F						
			*Rh. turanicus*	1N, 1M, 2F	1N, 1M, 1F						
	Goats	5/7	*Hy. anatolicum*	1N, 2M, 3F	1N, 1M, 1F						
			*Rh. haemaphysaloides*	2N, 2M, 4F	1N, 1M, 1F				1N	1N	1N
	Camels	2/3	*Hy. dromedarii*	2N, 2M, 3F	1N, 1F						
	Total	28/39		62	26	1					
Bannu	Equids	0/2	None	None	None						
	Cattle	7/7	*Hy. anatolicum*	1N, 2M, 3F	1N, 1M, 1F						
			*Rh. haemaphysaloides*	3N, 1M, 5F	1N, 1M, 1F						
			*Rh. microplus*	3N, 2M, 5F	1N, 1M, 1F						
	Buffaloes	4/6	*Rh. turanicus*	1N, 1M, 2F	1N, 1M, 1F				1N	1N	1N
	Sheep	3/9	*Hy. anatolicum*	1N, 2M	1N, 1M						
	Goats	5/7	*Hy. anatolicum*	1N, 2M, 2F	1N, 1M, 1F						
	Camels	2/4	*Hy. dromedarii*	2N, 1M, 2F	1N, 1M, 1F						
	Total	21/35		42	20	1					
Nowshera	Equids	2/2	*Rh. turanicus*	1N, 1M, 1F	1N, 1M, 1F						
	Cattle	8/9	*Hy. anatolicum*	1N, 2M, 3F	1N, 1M, 1F						
			*Rh. microplus*	4N, 2M, 6F	1N, 1M, 1F						
	Buffaloes	6/6	*Hy. anatolicum*	1N, 2M, 1F	1N, 1M, 1F						
			*Rh. turanicus*	1N, 1M, 1F	1N, 1F				1N	1N	1N
	Sheep	6/7	*Hy. anatolicum*	4N, 1M, 5F	1N, 1M, 1F						
	Goats	5/7	*Rh. haemaphysaloides*	4N, 1M, 6F	1N, 1M, 1F				1M	1M	
			*Ha. sulcata*	1N, 1M, 2F	1N, 1M, 1F						
	Camels	2/4	*Hy. dromedarii*	1N, 1M, 2F	1N, 1M, 1F						
	Total	29/35		57	26	2					
Bajaur	Equids	2/2	*Rh. turanicus*	1N, 2M, 1F	1N, 1M, 1F						
	Cattle	6/7	*Rh. turanicus*	1N, 2M, 2F	1N, 1M, 1F				1F	1F	
			*Rh. microplus*	5N, 1M, 5F	1N, 1M, 1F						
	Buffaloes	4/5	*Hy. anatolicum*	1N, 2M, 3F	1N, 1M, 1F						
			*Rh. turanicus*	1N, 1M, 1F	1N, 1M, 1F						
	Sheep	4/6	*Rh. turanicus*	1N, 1M, 2F	1N, 1M, 2F						
			*Rh. haemaphysaloides*	2N, 2M, 3F	2N, 1M, 1F				1F	1F	
	Goats	3/5	*Hy. anatolicum*	1N, 1M, 2F	1N, 1M, 1F						
			*Rh. haemaphysaloides*	1N, 2M, 2F	1N, 1M, 1F				1N	1N	1N
	Camels	1/2	*Hy. dromedarii*	1N, 1M, 2F	1N, 1F						
	Total	20/27		53	33	3					
Malakand	Equids	2/5	*Rh. turanicus*	1N, 1M, 2F	1N, 1F						
	Cattle	4/11	*Rh. turanicus*	1N, 1M, 2F	1N, 1M, 1F						
	Buffaloes	3/8	*Hy. anatolicum*	1N, 1M, 1F	1N, 1F						
	Sheep	4/9	*Ha. cornupunctata*	1N, 2M, 3F	1N, 2M,3F	1N	1N	1N			
	Goats	3/6	*Hy. anatolicum*	1N, 2M	1N, 1M						
	Camels	3/5	*Hy. dromedarii*	4N, 1M, 4F	1N, 1M						
	Total	19/44		29	18	1					
Mohmand	Equid	3/4	*Rh. turanicus*	1N, 1M, 3F	1N, 1M, 1F						
	Cattle	6/7	*Hy. anatolicum*	1N, 1M, 1F	1N, 1F						
			*Ha. cornupunctata*	4N, 1M, 4F	1N, 1M, 1F						
	Buffaloes	2/8	*Hy. anatolicum*	1N, 1M	1N, 1M						
	Sheep	5/10	*Rh. turanicus*	1N, 1M, 1F	1N, 1F				1N	1N	1N
			*Rh. haemaphysaloides*	1N, 2M	1N, 1M						
	Goats	4/6	*Rh. haemaphysaloides*	1N, 1M, 1F	1N, 1F				1F	1F	
			*Ha. sulcata*	1N, 1M	1N, 1M						
	Camels	3/4	*Hy. dromedarii*	3N, 1M, 3F	1N, 1M, 1F						
	Total	23/39		37	20	2					
Lower Dir	Equids	2/6	*Rh. turanicus*	1N, 1M, 1F	1M, 1F						
	Cattle	7/8	*Hy. anatolicum*	1N, 1F	1N, 1F						
			*Rh. microplus*	4N, 1M, 5F	2N, 1M, 2F						
	Buffaloes	4/6	*Rh. turanicus*	1N, 1M, 2F	1N, 1M, 1F				1F	1F	
	Sheep	7/10	*Ha. cornupunctata*	1N, 2M, 1F	1N, 1M, 1F						
			*Rh. turanicus*	1N, 1M, 2F	1N, 1F				1N	1N	1N
	Goats	6/9	*Ha. cornupunctata*	1N, 1M, 1F	1N, 1M, 1F						
			*Rh. haemaphysaloides*	1N, 1M, 2F	1N, 2F						
	Total	26/39		34	23	2					
Buner	Equids	3/4	*Rh. turanicus*	1N, 2M, 3F	1M, 1F						
	Cattle	6/7	*Rh. microplus*	3N, 2M, 4F	1N, 2F						
	Buffaloes	9/10	*Rh. turanicus*	1N, 1M, 1F	1N, 1F						
			*Hy. anatolicum*	1N, 1M, 1F	1N, 1M, 1F						
			*Rh. microplus*	4N, 2M, 5F	3N, 1F						
	Sheep	6/7	*Ha. sulcata*	1N, 1M, 1F	1N, 1M, 1F						
			*Rh. turanicus*	1N, 2M, 2F	1N, 1M, 1F						
	Goats	5/5	*Rh. turanicus*	1N, 1M, 1F	1N, 1M						
			*Rh. haemaphysaloides*	2N, 1M, 2F	1N, 1F				1F	1F	
	Camels	4/7	*Hy. dromedarii*	1N, 1M, 3F	1N, 1M, 1F						
	Total	33/40		53	27	1					
Swat	Equids	2/5	*Rh. turanicus*	1N, 1M, 1F	1N, 1F						
	Cattle	7/9	*Rh. microplus*	4N, 1M, 6F	1N, 1M, 1F						
	Buffaloes	8/10	*Rh. turanicus*	1N, 1M, 1F	1N, 1M, 1F						
			*Rh. microplus*	5N, 2M, 7F	1N, 1M,1F						
	Sheep	6/8	*Hy. anatolicum*	1N, 1M, 1F	1N, 1F						
			*Rh. turanicus*	1N, 2M, 2F	1N, 1M, 1F				1N	1N	1N
	Goats	6/7	*Hy. anatolicum*	1N, 1M, 2F	1N, 1F						
			*Ha. cornupunctata*	1N, 2M, 3F	1N, 1M, 1F						
	Total	29/39		49	19	1					
Total		373/575 (64.9%)	*Rh. turanicus* (163, 23.1%), *Rh. microplus* (179, 25.3%), *Hy. anatolicum* (136, 19.2%), *Rh. haemaphysaloides* (118, 16.6%), *Hy. dromedarii* (74, 10.5%), *Ha. sulcata* (9, 1.3%), *Ha. cornupunctata* (28, 1.8%)	707 (219N, 320F, 168M)	330 (118N, 95M, 117F)	4/330 (1.2%)			25/330 (7.6%)		
						25/330 (7.6%) *Rh. haemaphysaloides* (9/50, 18.0%) *Rh. turanicus* (13/99, 13.1%), *Rh. microplus* (2/49, 4.1%) *Ha. cornupunctata* (1/18, 5.5%),					

***N**: Nymph, **F**: Female, **M**: Male*.

## Data Availability

The data set of the current study can be found in the online repository under the accession numbers present in the article.

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
