# Peer review of "Association of SFG Rickettsia massiliae and Candidatus Rickettsia shennongii with Different Hard Ticks Infesting Livestock Hosts"

_pathogens, 2023, doi:10.3390/pathogens12091080_

Round 1
Reviewer 1 Report
Introduction:
1. Lines 64-72: Briefly describe findings in Pakistan's neighbouring countries as well.
2. Lines 75-81: briefly describe the findings, so that the basis for this study and the study area are clear for the reader.
Materials and Methods:
1. Study area: explain the rationale to choosing Khyber Pakhtunkhwa, and indicate that it is a Province.
2. Fig. 1. Indicate in the smaller map, the region amplified to the larger map, as well as the boundaries to Khyber Pakhtunkhwa for the readers not familiar with the geography of Pakistan.
3. Line 104: describe the method for random collection.
4. Line 106: indicate if the 1-8 ticks collected per animal were all ticks found or if 8 were the maximum collected even if there were more.
5. section 2.2.: also indicate the search method for each animal, i.e. if the entire body was examined, or only certain parts, and/or for how long.
6. Lines 124-125: indicate the buffer for the gel and the concentration of the ethidium bromide.
7. Lines 131: indicate which primers were used.
8. Line 136: indicate if gaps were present and how they were treated for phylogenetic analysis.
9. Lines 138-140: Why were all these phylogenetic methods used? - explain how this tests for "efficiency, consistency and robustness" - in particular because the results are not presented (for example on the trees to show if all methods produced the same result) or analyzed in the results section. Also, it's "maximum parsimony". But this sentence is inconsistent with the following sentence, which states "The Maximum likelihood method... used..." - which was it?
10. Line 140- indicate why the Tamura-Nei model was used. Did you use the model selection tool in MEGA?
Results:
1. Lines 147-151: These data should be presented on a table, for clarity, and summarized in the text, for example on Table 2, as the total for each district.
2. Lines 153-158: These data should be presented on a table, for clarity, and summarized in the text.
3. section 3.2.: indicate that the data are presented on table 2, from the start, not just at the end of the section. The data that is not presented on Table 2 should be included in a separate table.
4. Table 2: indicate as a footnote, what is M, F and M.
5. It should be clear, in the Results and Methods, if the authors identified Rickettsia massiliae or Candidatus Rickettsia shennongii for all the samples described in Table 2 by DNA sequencing. If so, then all sequences should be submitted to GenBank, not only a selection, even if they are indistinguishable. Whether or not there was intra-specific diversity is a result that must be reported, and discussed here.
6. section 3.3.: indicate coverage in the BLAST results, in addition to the % identity.
7. Figures 2-3: species for the samples obtained in this study should not be shown in the trees, as the trees are the evidence for the identification.
8. A phylogenetic analysis from concatenated sequences should be presented, particularly as many of the sequences included in the tree share the same GenBank code. If it is not possible to produce such a tree, then the authors should explain why. Ideally, the concatenated analysis should be the main tree, and the gene trees should be included as Supplementary Information, or as smaller figures, for example only showing in detail the Rickettsia massiliae or Candidatus Rickettsia shennongii groups and an outgroup.
Discussion:
1. Lines 217-219: This initial sentence makes little sense, as the manuscript is not about changes in tick or Rickettsia prevalence or distribution, but rather, as pointed out in the Introduction, providing data from a poorly sampled region.
Fig. 1: correct spellings on the legend to the colour codes (see: "Northren", "Southren", "Westren" and "boundry")
2. correct: line 146 ("animals consisted of"), line 152 ("belonging to three genera"), line 160 ("genomic DNA from 330"), line 181 (rickettsial not in italics)
3. The legends to figures 2-4 should be improved to a more standard format.
Author Response
INTRODUCTION:
- Lines 64-72: Briefly describe findings in Pakistan's neighboring countries as well.
Author’s response: Thank you. Suggestion accepted and updated accordingly in revised MS (lines 77-79).
- Lines 75-81: briefly describe the findings, so that the basis for this study and the study area are clear for the reader.
Author’s response: Thank you. Suggestion accepted. Modifications were inserted in the revised MS (lines 92-94).
MATERIALS AND METHODS:
- Study area: explain the rationale to choosing Khyber Pakhtunkhwa, and indicate that it is a Province.
Author’s response: Thank you. Suggestions accepted. Modifications were inserted in the revised MS (lines 103-109).
- 1. Indicate in the smaller map, the region amplified to the larger map, as well as the boundaries to Khyber Pakhtunkhwa for the readers not familiar with the geography of Pakistan.
Author’s response: Thank you. Suggestions accepted. Modifications were inserted in the revised map (Figure 1).
- Line 104: describe the method for random collection.
Author’s response: Thank you. Randomly means we collected ticks from hosts as where (within study area) we found these host animals during July 2019 to October 2020, we examined them to collect the ticks. We have not selected any specified hosts for repeated screening of them or interval of time for the tick’s collection.
- Line 106: indicate if the 1-8 ticks collected per animal were all ticks found or if 8 were the maximum collected even if there were more.
Author’s response: Thank you. As ticks were collected 1-8 per animals, 1 was the minimum number of tick collected from any animal, while 8 were the maximum number of ticks to be collected from animal. Some animals had 100s of ticks but we did not collected more than 8 ticks from the same animal’s body. We revised it as “only 1-8” to clarify the collection. (lines 123-125)
- Section 2.2.: also indicate the search method for each animal, i.e. if the entire body was examined, or only certain parts, and/or for how long.
Author’s response: Thank you. Modified that specific sentence. (Lines 123-125)
- Lines 124-125: indicate the buffer for the gel and the concentration of the ethidium bromide.
Author’s response: Thank you. Modified accordingly. (lines 142-145)
- Lines 131: indicate which primers were used.
Author’s response: All the three primers have been mentioned in Table 1.
- Line 136: indicate if gaps were present and how they were treated for phylogenetic analysis.
Author’s response: Thank you. No gaps were found in the obtained sequences during analysis. Both gltA, ompA and ompB are protein coding sequences, so it was expecting that they did not present any gap with closest sequences in GenBank. Only start and the end region of the obtained sequences were trimmed via SeqMan. (line 150)
- Lines 138-140: Why were all these phylogenetic methods used? - Explain how this tests for "efficiency, consistency and robustness" - in particular because the results are not presented (for example on the trees to show if all methods produced the same result) or analyzed in the results section. Also, it's "maximum parsimony". But this sentence is inconsistent with the following sentence, which states "The Maximum likelihood method... used..." - which was it?
Author’s response: Thank you for highlighting this part. All methods were not producing the same results, hence we tested all the methods (Maximum likelihood, Neighbor-Joining, Minimum-Evolution, Parsimony, and UPGMA), and finalized the “Maximum likelihood method (Tamura-Nei model)” for Phylogenetic analysis through which we obtained good node values and correct topology. Correction has been made in the above highlighted lines (158-161)).
- Line 140- indicate why the Tamura-Nei model was used. Did you use the model selection tool in MEGA?
Author’s response: Thank you for suggestions. As we have used all the methods (Maximum likelihood, Neighbor-Joining, Minimum-Evolution, Parsimony, and UPGMA) along with each model (p-distance, Jukes cantor model, Kimura 2 parameter model, Tajima Nei model, Tamura 3 parameter model, Tamura Nei model, and Maximum composite likelihood), and we found the best node values as well a correct tree topology only by applying “Maximum likelihood along with Tamura Nei model” in this MS. Hence we selected to use these models.
RESULTS:
- Lines 147-151: These data should be presented on a table, for clarity, and summarized in the text, for example on Table 2, as the total for each district.
Author’s response: Thank you. Suggestions followed. (section 3.1)
- Lines 153-158: These data should be presented on a table, for clarity, and summarized in the text.
Author’s response: Thank you. Suggestions followed. (section 3.1)
- Section 3.2.: indicate that the data are presented on table 2, from the start, not just at the end of the section. The data that is not presented on Table 2 should be included in a separate table.
Author’s response: Thank you. Suggestions followed, as all the data have been mentioned in Table 2. (Section 3.2, lines 185-187)
- Table 2: indicate as a footnote, what is M, F and M.
Author’s response: Thank you. Suggestions followed. (line 201)
- It should be clear, in the Results and Methods, if the authors identified Rickettsia massiliae or Candidatus Rickettsia shennongii for all the samples described in Table 2 by DNA sequencing. If so, then all sequences should be submitted to GenBank, not only a selection, even if they are indistinguishable. Whether or not there was intra-specific diversity is a result that must be reported, and discussed here.
Author’s response: Yes, all tick samples presented in Table 2 were submitted to DNA sequence and two Rickettsia species were identified: Rickettsia massiliae and ‘Candidatus Rickettsia shennongii’. We have added to the text that only one haplotype was generated for each gene of ‘Ca. R. shennongii’ or R. massiliae (lines 204-220). For this reason, this single haplotype was used in each of the phylogenetic analyses (gltA, ompA, and ompB). At the end, we had a total of 6 distinct haplotypes in this study. For submission, GenBank asked to give a name to each haplotype for each rickettsial gene. Then, we submitted the six haplotypes to GenBank: two gltA (R. massiliae and ‘Ca. R. shennongii’), two ompA (R. massiliae and ‘Ca. R. shennongii’), and two ompB (R. massiliae and ‘Ca. R. shennongii’). Although this reviewer has asked us to submit all DNA sequences from each tick specimen to GenBank, we would like to maintain the submission of only one copy of each haplotype, to avoid redundant sequences to GenBank that would not make any improvement to our manuscript.
- Section 3.3.: indicate coverage in the BLAST results, in addition to the % identity.
Author’s response: Suggestion accepted as mentioned queries at their specific section (3.3). (lines 210-214)
- Figures 2-3: species for the samples obtained in this study should not be shown in the trees, as the trees are the evidence for the identification.
Author’s response: Thank you. Suggestion accepted and followed.
- A phylogenetic analysis from concatenated sequences should be presented, particularly as many of the sequences included in the tree share the same GenBank code. If it is not possible to produce such a tree, then the authors should explain why. Ideally, the concatenated analysis should be the main tree, and the gene trees should be included as Supplementary Information, or as smaller figures, for example only showing in detail the Rickettsia massiliae or Candidatus Rickettsia shennongii groups and an outgroup.
Author’s response: Thanks for constructive suggestions. For constructing the concatenated phylogenetic tree, only sequences from the same strain or complete genomes should be used. Initially we have tried to construct a concatenated tree based on all genetic markers (gltA, ompA, and ompB), however, due to the lack of any of these three marker sequences in NCBI for some Rickettsia spp., we opted to construct separate phylogenetic tree for each genetic marker.
Similarly, we then searched the full genome sequences for Candidatus Rickettsia shennongii and other Rickettsia spp. to obtain the sequences for these three genes, but this data was also very limited to construct a good topology concatenated phylogenetic tree.
Additionally, full genome sequences for some species were available in the GenBank but our one or two genetic markers were not hitting the aligned regions in that complete genome. For Example, Rickettsia aeschlimannii full genome with accession number (CCER01000003.1) was available but our obtained ompA sequence was not hitting the aligned region in that complete genome of Rickettsia aeschlimannii.
Full genome for Rickettsia sp. (EF219463, MT434984, EF219467, MT434986, and EF219464) and Ca. Rickettsia shennongii (OL856117, OL856116, OL856104, and ON015829) were not available in the GenBank. Therefore, we opted to finalize the individual tree for each genetic marker. Please note that even presenting three independent trees, they were all congruent in indicating the same results, i.e., our new haplotypes grouped with R. massiliae and ‘Ca. Rickettsia shennongii’.
DISCUSSION:
- Lines 217-219: This initial sentence makes little sense, as the manuscript is not about changes in tick or Rickettsia prevalence or distribution, but rather, as pointed out in the Introduction, providing data from a poorly sampled region.
Author’s response: Thank you. Modified accordingly. (lines 257-284)
Comments on the Quality of English Language
Fig. 1: correct spellings on the legend to the colour codes (see: "Northren", "Southren", "Westren" and "boundry")
Author’s response: Thank you for spotting this. Corrected accordingly.
- correct: line 146 ("animals consisted of"), line 152 ("belonging to three genera"), line 160 ("genomic DNA from 330"), line 181 (rickettsial not in italics)
Author’s response: Thank you. Corrections were made in the revised MS. (lines 165, 176)
- The legends to figures 2-4 should be improved to a more standard format.
Author’s response: Suggestions followed. (figure 2-4)
Reviewer 2 Report
Dear Authors,
Your study presents the detection of two different Rickettsia spp. detected in ticks collected on livestock from Pakistan. The used methods are adequate for the purpose of the study. The results are properly presented and analyzed. Also the discussion is well-founded. Please find attached my comments and corrections:
Lines 21/22: I do not agree with this statement. There is a lot of information about the genetic characterization and epidemiology of Rickettsia spp. If you are talking about the special case of Pakistan, please change the sentence and clarify this point.
Lines 28-31: You say that the overall prevalence of Rickettsia spp. was 7.6%; 25 positive samples in 330 total samples. Afterward you declare the positive samples, but when you sum up the total samples of the four tick species you just come to 216 samples. Please explain this difference in the sample numbers
Line 33: Please change `Candidatus (Ca) Rickettsia (R) shennongii` to Candidatus Rickettsia shennongii
Line 36: delete ` ` in candidatus names
Lines 48-49: Restructure this sentence. The fact that these bacteria may infect humans does not lead to significant economic losses.
Lines 61-62: Please order the species names in alphabetic order. The last “and” without italic.
Line 89: Please explain that Khyber Pakhtunkhwa is a province of Pakistan
Lines 103-110: The collected ticks were attached on the samples animals? Or were they free-walking on the host? This is important as the detection of rickettsial DNA in the tick may be a result of ingesting infected host blood. Please add this information
Line 114: It is Table 2 not 1. Please change the numbers of the tables as table 2 is named before table 1 in the text.
Line 126: Did you add the ethidium bromide to the gel or did you prepare a bath after the electrophoresis? Please clarify.
Lines 160-172: Please present these results more detailed: All positive samples were positive in the three PCR assays? Or were there some samples that just show positive results in one or two PCRs?
Lines 166-167: These tick species were named previously so that you must use abbreviations in the genus name.
Line 177 and 190: Please change `Candidatus (Ca) Rickettsia (R) shennongii` to Ca. R. shennongii
Lines 190: Which sequences did you deposited in the GenBank? Tick species? Host? Sample locality? Please add this information
Lines 197-199: Please add the length of the alignment.
Lines 203-205: Please add the length of the alignment. Have you any explication for the separation of the R. massiliae clade in two branches? Please discuss this.
Lines 209-214: Please add the length of the alignment.
Lines 216 ff: Discussion. Please check the spelling and abbreviations of the species names
Lines 223-234: These are results and should be moved to the “Results” section.
Lines 249-251: R. massiliae also was detected in Argentina (see. Cicuttin et al. 2014, 2015; Monje et al. 2016)
Lines 259, 267, 269, 274, 276, 277, 281, 285, 296, 301: Please change `Ca. R. shennongii` to Ca. R. shennongii
Minor editing of English language required
Author Response
Your study presents the detection of two different Rickettsia spp. detected in ticks collected on livestock from Pakistan. The used methods are adequate for the purpose of the study. The results are properly presented and analyzed. Also the discussion is well-founded. Please find attached my comments and corrections:
- Lines 21/22: I do not agree with this statement. There is a lot of information about the genetic characterization and epidemiology of Rickettsia If you are talking about the special case of Pakistan, please change the sentence and clarify this point.
Author’s response: Thank you. We agree that there is a lot of information on Rickettsia spp. from Pakistan however, most of the genetic characterization of these Rickettsia spp. are either based on one or two genetic markers (mostly short fragments of 16s and gltA). Since gltA and 16s are highly conserved among different Rickettsia spp., thus, these genetic markers do not delineate the species with accuracy. Here we used three genetic markers (gltA, ompA, ompB) that have been used and recommended suitable for genetic characterization of different Rickettsia spp. Therefore, we stated “Information are barely available about the “genetic characterization” and epidemiology of Rickettsia spp.
- Lines 28-31: You say that the overall prevalence of Rickettsia was 7.6%; 25 positive samples in 330 total samples. Afterward you declare the positive samples, but when you sum up the total samples of the four tick species you just come to 216 samples. Please explain this difference in the sample numbers.
Author’s response: Thank you. In abstract of this MS, we have only mentioned those tick species which were found positive for Rickettsia spp. such as Rh. haemaphysaloides, Rh. turanicus, Rh. microplus and Ha. cornupunctata. The total number of these four tested ticks was 216. This calculation (total 330) is clear in Table 2 of result section. Now we also added a sentences to clarify it further and mentioned tick species such as Hy. anatolicum, Hy. dromedarii, and Ha. sulcata that were negative for Rickettsia spp. (lines 33-34)
Line 33: Please change `Candidatus (Ca) Rickettsia (R) shennongii` to Candidatus Rickettsia shennongii
Author’s response: Thank you. Modified accordingly. (line 36)
- Line 36: delete ` ` in candidatus names
Author’s response: Thank you. Modified accordingly. (line 36, and throughout MS)
- Lines 48-49: Restructure this sentence. The fact that these bacteria may infect humans does not lead to significant economic losses.
Author’s response: Thank you. Modified that statement. (lines 51-54)
- Lines 61-62: Please order the species names in alphabetic order. The last “and” without italic.
Author’s response: Thank you. Modified accordingly. (lines 65-68)
- Line 89: Please explain that Khyber Pakhtunkhwa is a province of Pakistan
Author’s response: Thank you. Modified accordingly. (line 98)
- Lines 103-110: The collected ticks were attached on the samples animals? Or were they free-walking on the host? This is important as the detection of rickettsial DNA in the tick may be a result of ingesting infected host blood. Please add this information
Author’s response: Thank you. Modified the statement accordingly as attached ticks were collected from hosts. Modification were inserted accordingly in discussion section of revised MS. (lines 119-122)
- Line 114: It is Table 2 not 1. Please change the numbers of the tables as table 2 is named before table 1 in the text.
Author’s response: Thanks a lot for highlighting this mistake. Modified accordingly. (line 130)
- Line 126: Did you add the ethidium bromide to the gel or did you prepare a bath after the electrophoresis? Please clarify.
Author’s response: Thank you. “Ethidium bromide” was added to the gel. As commented by other reviewer, concentrations were also added. (line 140-142)
- Lines 160-172: Please present these results more detailed: All positive samples were positive in the three PCR assays? Or were there some samples that just show positive results in one or two PCRs?
Author’s response: Thank you for your concern on this part. As mentioned in details in Table 2, that 330 ticks were individually tested for each gltA, ompA, and ompB. Positive samples were either positive for combination of either two or three genes or alone for ompB gene. ompB positive PCR were sequenced and consider positive as this DNA fragment has sufficient size and highly mutated and can differentiate Rickettsia spp.
- Lines 166-167: These tick species were named previously so that you must use abbreviations in the genus name.
Author’s response: Thank you. Modified accordingly. (lines 193-194)
- Line 177 and 190: Please change `Candidatus (Ca) Rickettsia (R) shennongii` to R. shennongii
Author’s response: Thank you. Modified accordingly. (Lines 205)
- Lines 190: Which sequences did you deposited in the GenBank? Tick species? Host? Sample locality? Please add this information
Author’s response: We have mentioned that “the obtained haplotypes of Ca. R. shennongii (gltA, ompA, ompB) and R. massiliae (gltA, ompA, ompB) sequences were deposited to GenBank. Including all detailed data about host of Rickettsia spp., organism, and country
- Lines 197-199: Please add the length of the alignment.
Author’s response: Thank you. Modified accordingly. (figure 2)
- Lines 203-205: Please add the length of the alignment. Have you any explication for the separation of the massiliae clade in two branches? Please discuss this.
Author’s response: Thank you. Modified accordingly. (figure 3)
- Lines 209-214: Please add the length of the alignment.
Author’s response: Thank you. Modified accordingly. (figure 4)
- Lines 216 ff: Discussion. Please check the spelling and abbreviations of the species names
Author’s response: Thank you. Checked.
- Lines 223-234: These are results and should be moved to the “Results” section.
Author’s response: Thank you. All these have been mentioned in “Results section 3.2, 3.3. hence, repetition has been removed from discussion section.
- Lines 249-251: massiliae also was detected in Argentina (see. Cicuttin et al. 2014, 2015; Monje et al. 2016)
Author’s response: Thank you. Modified accordingly. (line 287)
- Lines 259, 267, 269, 274, 276, 277, 281, 285, 296, 301: Please change `R. shennongii` to Ca. R. shennongii
Author’s response: Thank you. Modified accordingly.
Round 2
Reviewer 1 Report
I am happy with many changes and replies, but not all.
1. Line 113 - if the authors are going to state that samples were "randomly collected", then they must provide information of how randomness was achieved. What was the field work strategy followed for sampling? Did the authors go to farms, or open fields, or along roads, around villages? Did they follow a grid?
2. I do insist that all sequences must be submitted to GenBank, following from the requirement that original data must be made available to other researchers. An example does not follow that principle. Researchers should be able to access all sequences obtained in this work.
3. The authors should explain what they mean by "correct topology" - what measure did they use to determine what is "correct", particularly when considering that different methods resulted in different trees. Whereas Maximum Likelihood is usually considered the best method (of those available in MEGA) for phylogenetic reconstruction, the best mutation model can vary widely depending on the data. You can run the "Find best DNA/protein models" under the tab "Models" to choose the model. Alternatively, or additionally, indicate the likelihood values for different ML trees under different models to justify the choice of the model.
4. Line 150: "were used to test for the efficiency, consistency, and robustness". It is still not clear what is meant by this. Explain what you mean by each of these terms in the manuscript, how the evaluation was done and how was the best method selected- what were the criteria.
5. Lines 194-196: "For each of these two Rickettsia species, a single haplotype was generated for each of the three rickettsial genes, regardless of the number of PCR-positive ticks." This sentence is unclear. It seems to mean that only one sequence was obtained per species. If all products were sequenced, be clear about this. If the obtained sequences were all identical within each species, and only one was used in the phylogenetic analysis, then also be clear about this.
Overall the quality of English is sufficient, but it should be reviewed carefully.
Author Response
Reviewer # 1, comments
- Line 113 - if the authors are going to state that samples were "randomly collected", then they must provide information of how randomness was achieved. What was the field work strategy followed for sampling? Did the authors go to farms, or open fields, or along roads, around villages? Did they follow a grid?
Author’s response: Thank you. Modification has been done in the highlighted part. (line 113-118). Since we collected ticks only when we detected tick-infested hosts, we replaced the term "randomly collected" by “conveniently collected”.
I do insist that all sequences must be submitted to GenBank, following from the requirement that original data must be made available to other researchers. An example does not follow that principle. Researchers should be able to access all sequences obtained in this work.
Author’s response: We still do not understand this suggestion and would like to ask the Editor to advise us. Why should we submit repeated times to GenBank the same DNA sequence from the same study? For example, some of us recently published a manuscript in Pathogens (Pathogens 2023, 12, 446. https://doi.org/10.3390/pathogens12030446), in which “Candidatus Rickettsia andeanae” was detected in 21 tick specimens; because a single haplotype of gltA and a single haplotype of ompA were generated from these 21 tick specimens, only one submission of each gene was sent to GenBank. We could provide here many other examples of such procedure, which have been widely accepted to avoid redundancies in GenBank. Of course, the opposite has also been done -i.e, multiple submissions of the same haplotype from the same study to GenBank-; however, indeed this redundant procedure has been the exception. For this reason, we would like to ask the editor to accept our procedure to submit each haplotype only once to GenBank, following the majority of the studies that have been published.
- The authors should explain what they mean by "correct topology" - what measure did they use to determine what is "correct", particularly when considering that different methods resulted in different trees. Whereas Maximum Likelihood is usually considered the best method (of those available in MEGA) for phylogenetic reconstruction, the best mutation model can vary widely depending on the data. You can run the "Find best DNA/protein models" under the tab "Models" to choose the model. Alternatively, or additionally, indicate the likelihood values for different ML trees under different models to justify the choice of the model.
Author’s response: Thank you for the comment. The phylogenetic trees were constructed based on different available methods such as Maximum likelihood, Neighbor-Joining, Minimum-Evolution, Parsimony, and UPGMA in MEGA-X and compared all the constructed phylogenetic trees. All these methods and their different models were tested, being found similar results. However, the maximum likelihood has been commonly used method for the evolutionary analysis of ticks and tick-borne pathogens, due to its ability to evaluate different phylogenetic trees and models under a statistical framework (Guindon et al: https://doi.org/10.1080/10635150390235520 and https://doi.org/10.1007/978-1-59745-251-9_6). To avoid the confusion, we specified the method and its model for the phylogenetic trees in the revised MS. Moreover, the topology of Rickettsia spp. in this MS was in accordance to Karkouri et al., 2022 (https://doi.org/10.1038/s41598-022-07725-z), as maximum likelihood is used by Karkouri et al., 2022.
- Line 150: "were used to test for the efficiency, consistency, and robustness". It is still not clear what is meant by this. Explain what you mean by each of these terms in the manuscript, how the evaluation was done and how was the best method selected- what were the criteria.
Author’s response: Thank you. The methodology sections “2.4. DNA Sequencing and Phylogenetic Analysis” were modified. The phylogenetic trees were constructed based on different available methods such as Maximum likelihood, Neighbor-Joining, Minimum-Evolution, Parsimony, and UPGMA in MEGA-X and compared all the constructed phylogenetic trees. All these methods and their different models were tested, being found similar results. However, the maximum likelihood is a recommended and accurate method for the best evolutionary analysis, due to its ability to evaluate different phylogenetic trees and models under a statistical framework (Guindon et al: https://doi.org/10.1080/10635150390235520 and https://doi.org/10.1007/978-1-59745-251-9_6). To avoid the confusion, we specified the method and its model for the phylogenetic trees in the revised MS. Moreover, the topology of Rickettsia spp. in this MS was in accordance to Karkouri et al., 2022 (https://doi.org/10.1038/s41598-022-07725-z), as maximum likelihood is used by Karkouri et al., 2022.
- Lines 194-196: "For each of these two Rickettsia species, a single haplotype was generated for each of the three rickettsial genes, regardless of the number of PCR-positive ticks." This sentence is unclear. It seems to mean that only one sequence was obtained per species. If all products were sequenced, be clear about this. If the obtained sequences were all identical within each species, and only one was used in the phylogenetic analysis, then also be clear about this.
Author’s response: Yes, DNA sequences were generated from all 25 tick specimens that yielded amplicons by PCR assays targeting fragments of the rickettsial genes gltA, ompA, and ompB. As suggested by this reviewer, we have made it clearer in the text. Also, as we mention in the text, for each rickettsial gene a single haplotype was generated for ‘Candidatus Rickettsia shennongii’ and a single haplotype was generated for R. massiliae, regardless of the number of PCR-positive ticks. For this reason, only one haplotype of each gene was used in the phylogenetic analyses and was submitted to GenBank, obviously to avoid redundancy. Thank you.
Reviewer 2 Report
Dear authors,
thank you for the accurate revision of the manuscript.
Round 3
Reviewer 1 Report
Thank you for addressing most of my concerns, in particular regarding data availability for the scientific community.
It can be accepted now.
However, I would still recommend that it is explained in the manuscript why the Tamura-Nei model was used (models should be chosen that best match the data - and MEGA has a tool to do this - and it should be used in future studies, if not done now).
It is also useful to state that methods other than ML were used and that the topology was similar, and that the topology recovered here was similar to other published ones because it reinforces the quality of the trees presented here.